# Biomarkers of Frailty in Patients with Advanced Chronic Liver Disease Undergoing a Multifactorial Intervention Consisting of Home Exercise, Branched-Chain Amino Acids, and Probiotics

**DOI:** 10.3390/biom14111410

**Published:** 2024-11-06

**Authors:** Luca Laghi, Maria Àngels Ortiz, Giacomo Rossi, Eva Román, Carlo Mengucci, Elisabet Cantó, Lucia Biagini, Elisabet Sánchez, Maria Mulet, Álvaro García-Osuna, Eulàlia Urgell, Naujot Kaur, Maria Poca, Josep Padrós, Maria Josep Nadal, Berta Cuyàs, Edilmar Alvarado, Silvia Vidal, Elena Juanes, Andreu Ferrero-Gregori, Àngels Escorsell, German Soriano

**Affiliations:** 1Department of Agricultural and Food Sciences, University of Bologna, 47521 Cesena, Italy; carlo.mengucci2@unibo.it; 2Institut de Recerca Sant Pau (IR Sant Pau), 08041 Barcelona, Spain; mortiz@santpau.cat (M.À.O.); ecanto@santpau.cat (E.C.); esanchezar@santpau.cat (E.S.); mmulet@santpau.cat (M.M.); svidal@santpau.cat (S.V.); andreu.ferrero.gregori@gmail.com (A.F.-G.); 3School of Veterinary Medical Sciences, University of Camerino, 62032 Camerino, Italy; giacomo.rossi@unicam.it (G.R.); lucia.biagini@unicam.it (L.B.); 4CIBERehd (Centro de Investigación Biomédica en Red de Enfermedades Hepáticas y Digestivas), Instituto de Salud Carlos III, 28029 Madrid, Spain; eroman@santpau.cat (E.R.); mpoca@santpau.cat (M.P.); bcuyas@santpau.cat (B.C.); ealvaradot@santpau.cat (E.A.); 5University Nursing School EUI-Sant Pau, 08025 Barcelona, Spain; 6Department of Gastroenterology, Hospital de la Santa Creu i Sant Pau, 08041 Barcelona, Spain; nkaur@santpau.cat (N.K.); mescorsell@santpau.cat (À.E.); 7Department of Biochemistry, Hospital de la Santa Creu i Sant Pau, 08041 Barcelona, Spain; agarciao@santpau.cat (Á.G.-O.); eurgell@santpau.cat (E.U.); 8Department of Physical Medicine and Rehabilitation, Hospital de la Santa Creu i Sant Pau, 08041 Barcelona, Spain; jpadros@santpau.cat (J.P.); mnadalc@santpau.cat (M.J.N.); 9Department of Cellular Biology, Physiology and Immunology, Universitat Autònoma de Barcelona, 08193 Bellaterra, Spain; 10Department of Pharmacy, Hospital de la Santa Creu i Sant Pau, 08041 Barcelona, Spain; ejuanes@santpau.cat; 11Faculty of Medicine, Universitat Autònoma de Barcelona, 08193 Bellaterra, Spain

**Keywords:** metabolomics, exercise, branched-chain amino acids, probiotics, inflammation

## Abstract

Frailty in cirrhosis or advanced chronic liver disease (ACLD) is a relevant prognostic factor. In the present study, we aimed to analyze potential biomarkers associated with frailty and its improvement in patients with ACLD. We analyzed the serum of outpatients with ACLD who participated in a previous study (Román, Hepatol Commun 2024) in which frailty was assessed using the liver frailty index (LFI), and patients who were frail or prefrail were randomized to a multifactorial intervention (home exercise, branched-chain amino acids, and probiotics) or control for 12 months. We determined a biomarker battery of inflammation, bacterial translocation, and liver damage in blood and urine and blood metabolomics by ^1^H-NMR. Thirty-seven patients were included. According to the LFI, 32 patients were frail or prefrail, and 5 were robust. At baseline, LFI correlated with LBP, sCD163, mtDNA, FGF-21, urinary NGAL, urinary claudin-3, and the metabolites mannose, ethanol, and isoleucine. During the study, patients in the intervention group showed an improvement in LFI and a decrease in CRP, LBP, sCD163, and ccK18 compared to the control group. Metabolomics showed a decrease in dimethyl sulfone and creatinine and an increase in malonate, ornithine, isoleucine, and valine in the intervention group. We conclude that frailty in patients with ACLD is associated with biomarkers of systemic inflammation, bacterial translocation, and liver damage, and alterations of amino acid and short-chain fatty acid metabolism.

## 1. Introduction

Frailty is defined as a syndrome consisting of a decrease in homeostasis, reserve capacity, and resistance to stress that favors the occurrence of adverse outcomes, such as disability, hospitalization, and premature death [1,2]. The frailty syndrome is considered multidimensional because it includes not only sarcopenia—the decrease in muscle mass and/or function—but also a variety of other factors, such as physical limitation, mood alterations, cognitive impairment, and social aspects [1,2,3].

The frailty syndrome was first described in older people [1,2], but it has also been recently associated with chronic diseases [3], including cirrhosis, independently of age [4,5,6]. In patients with cirrhosis or advanced chronic liver disease (ACLD), and independently of the degree of liver failure, frailty has been shown to be a prognostic factor for mortality, higher healthcare needs, and poorer health-related quality of life [4,5,6].

Because of its clinical relevance, frailty has been identified as a therapeutic target in patients with ACLD, and several non-pharmacological strategies to ameliorate this condition have been proposed [4], such as exercise [7,8,9,10], nutritional interventions [10,11,12], branched-chain amino acid (BCAA) supplements [10,12,13,14], testosterone supplementation in men [15], mineral-rich, low sodium water [16], and probiotics [17]. Regarding exercise, patients with chronic diseases tend to perform less physical activity [18], as do patients with ACLD, in which exercise programs have shown benefit in physical function and sarcopenia [7,8,9,10].

We recently observed that a non-pharmacological, multifactorial intervention for patients with cirrhosis consisting of home exercise, BCAA, and a multistrain probiotic improved frailty and reduced emergency room consultations and falls [19].

Identifying biomarkers related to frailty has recently prompted interest in order to increase our understanding of the physiopathological mechanisms involved and to identify new therapeutic measures. In recent years, several biomarkers associated with prognosis and frailty have been described in patients with cirrhosis, most of which have been related to inflammation, bacterial translocation, and impaired metabolic pathways [6,17,20,21,22,23].

The aim of the present study was to explore potential biomarkers associated with frailty and its improvement after the above-mentioned multifactorial intervention in patients with ACLD.

## 2. Patients and Methods

### 2.1. Study Design and Patient Selection

We analyzed serum samples from outpatients with cirrhosis included in a recently published study [19]. All patients were aged >18 years old, and ACLD was diagnosed by clinical, analytical, and ultrasonographic criteria, or liver biopsy. Exclusion criteria were patients with poor prognosis (expected survival < 6 months), hepatocellular carcinoma or other active neoplastic disease, overt hepatic encephalopathy, neurological disorder, alcohol consumption in the previous 3 months, severe comorbidities, hospitalization in the previous month, contraindications to exercise or probiotic treatment, and refusal to give informed consent.

Frailty was assessed by the liver frailty index (LFI) [4,24]. Patients who were frail or prefrail were randomized to either the multifactorial intervention including home exercise (3 sessions/week of 20–30 min, progressively increasing according to the degree of tolerance to 40–60 min), BCAA (leucine, isoleucine, and valine, 10 g 30 min before each exercise session) and a multistrain probiotic (Vivomixx^®^, Mendes SA, Lugano, Switzerland, one sachet of 450 × 10^9^ bacteria every 12 h), or control for 12 months. LFI and body composition by bioelectrical impedance analysis (BIA) and ultrasound as parameters of frailty [19] were assessed at baseline and every 3 months. The LFI includes the evaluation of handgrip strength, the ability to get up from and sit on a chair (timed chair stands), and balance, and is currently the most widely accepted tool to evaluate frailty in patients with cirrhosis. Detailed instructions to perform the LFI can be found at https://liverfrailtyindex.ucsf.edu/, accessed on 12 February 2020. Information regarding LFI, BIA and ultrasound performance and the multifactorial intervention program is detailed in Appendix A.

At baseline and every 3 months for 12 months, blood samples were drawn and stored at −80 °C to later determine a battery of biomarkers that we considered relevant in the context of frailty in cirrhosis [6,17,19,20,21,22,23]. This battery included c-reactive protein (CRP), TNF-α, IL-6, tumor necrosis factor soluble receptor 1 (TNFsR1), IgA, copeptin, leucine rich alpha-2-glycoprotein 1 (LRG1), lipopolysaccharide binding protein (LBP), soluble CD163 (sCD163), soluble mannose receptor (sMR), caspase-cleaved keratin 18 (ccK18), mitochondrial DNA (mtDNA), cystatin C, myostatin, resistin, vitamin D, alpha-2-Heremans-Schmid glycoprotein (AHSG), growth differentiation factor 15 (GDF-15), fibroblast growth factor 21 (FGF-21), malondialdehyde (MDA), and blood metabolomics by ^1^H-NMR. At baseline, urine samples to determine claudin-3 and neutrophil gelatinase-associated lipocalin (uNGAL) were obtained in a subgroup of patients and stored at −80 °C. The clinical meaning of the battery of biomarkers is shown in Appendix A.

The study was conducted according to guidelines of the Declaration of Helsinki and approved by the Ethics Committee of Hospital de la Santa Creu i Sant Pau (Comitè d’Ètica d’Investigació amb Medicaments-CEIM) on 23 August 2019, approval number IIBSP-FRA-2019-36, 19/212. The protocol was registered at ClinicalTrials.gov (NCT01686698).

### 2.2. Methods for Biomarkers Analysis

Resistin, myostatin, TNF-α, IL-6, TNFsR1, IgA, copeptin, LRG1, AHSG, FGF-21, and GDF-15 were measured by ELISA in plasma, while claudin-3 and uNGAL were measured by ELISA in urine. Commercial kits were used for all measurements according to the manufacturers’ protocols. Kits for resistin, myostatin, TNFsR1, AHSG, FGF-21, GDF-15, and uNGAL were obtained from R&D Systems (Minneapolis, MN, USA); IL-6 from ImmunoTools (Friesoythe, Germany); TNF-α and IgA from Invitrogen (Waltham, MA, USA); claudin-3, copeptin, and LRG1 from Elabscience (Houston, TX, USA). Serum concentrations of CRP and cystatin C were measured using automated immunoturbidimetric assays on the Abbott Alinity^®^ c platform (Abbott Laboratories, Chicago, IL, USA). 25-OH vitamin D was measured using a chemiluminescent immunoassay of microparticles on the Abbott Alinity^®^ platform (Abbott Laboratories, Chicago, IL, USA). MDA in plasma was determined using a spectrophotometric method detailed in Appendix A.

LBP was measured by means of a sandwich assay with an immunometric chemiluminescent substrate using an automated analyzing system (Immulite LBP, DPC, Los Angeles, CA, USA); sCD163 by Macro163 sandwich ELISA (Trillium Diagnostics, Bangor, ME, USA); sMR by ELISA test (catalog n° # HK381, Hycult Biotech, Uden, The Netherlands); and ccK18 by the M30-Apoptosense ELISA kit (VLVbio [Peviva], Nacka, Sweden).

For mtDNA analysis, cell-free DNAs were isolated using Norgen Plasma/Serum Cell-Free Circulating DNA Purification Kit (Catalog No. 55100). The concentration and purity of the isolated DNA were determined using a NanoDrop Spectrometer. The relative quantification of mtDNA copy number was determined using quantitative real-time PCR (qPCR), and absolute quantification of mtDNA copy number was performed using the QuantStudio 3D Digital PCR (dPCR) System. As an endogenous gene control, we used the nuclear low copy gene β-actin (GenBank accession number NM_001101).

### 2.3. Metabolomics by ^1^H-NMR

For the analysis of the serum metabolome using ^1^H-NMR spectroscopy, we followed our own protocol [25]. An NMR analysis solution was prepared containing 10 mmol/L of 3-(trimethylsilyl)-propionic-2,2,3,3-d_4_ acid sodium salt (TSP) in D2O, adjusted to a pH of 7.00 ± 0.02 using a 1 M phosphate buffer. This solution also included 10 μL of 2 mmol/L NaN_3_ to inhibit microbial growth. TSP served as the chemical shift reference for the NMR spectra. Serum sample preparation involved thawing and centrifuging 1 mL of each sample at 18630 g for 15 min at 4 °C. Following centrifugation, 0.7 mL of the supernatant were mixed with 0.1 mL of the NMR analysis solution. The resulting mixture was centrifuged again under the same conditions immediately before analysis.

^1^H-NMR spectra were obtained at 298 K using an AVANCE III spectrometer (Bruker, Milan, Italy), equipped with Topspin software (version 3.5-pl6) and operating at 600.13 MHz. To suppress broad signals from slow-tumbling molecules, we used a CPMG filter with a 330 ms duration, composed of 400 echoes generated by 180° pulses each lasting 24 μs and separated by 400 μs. Presaturation was applied to suppress the residual water signal using the cpmgpr1d sequence from the standard pulse sequence library. Each spectrum was acquired by summing 256 transients using 32 K data points over a 7184 Hz spectral window, with an acquisition time of 2.28 s and a recycle delay set at 5 s.

For untargeted quantification, signals with sufficient intensity were assigned by comparing chemical shift, multiplicity, and shape using Chenomx software (Chenomx Inc., Edmonton, AB, Canada, version 8.3) with its proprietary library (version 11) and the Human Metabolome Data Bank library (release 2). After phase and baseline corrections with the corresponding routines in Topspin, spectra were imported into the R computational environment. Molecules in the first analyzed sample were quantified using an external standard. Variations in water content among samples were adjusted using probabilistic quotient normalization [26]. Each molecule’s signals were integrated using rectangular integration.

### 2.4. Statistical Analysis

Data are expressed as frequencies, percentages, and mean ± SEM. Student’s *t*-test (if normal distribution) or Mann–Whitney test (if non-normal distribution) were used to compare quantitative variables between the two groups at baseline. Normality of data distribution was assessed using the Shapiro–Wilk test. Fisher’s test was used to compare qualitative variables. Pearson or Spearman tests were used for correlations. Multivariable linear regression analysis was performed to assess factors associated with LFI at baseline. To compare the overall evolution of the LFI and the biomarkers between the two groups throughout follow-up, we used linear mixed models, adjusting for baseline values. Metabolomic data were analyzed comparing the delta change at each time point (3, 6, 9, and 12 months) with respect to baseline results between the two groups using two-way ANOVA. A two-sided *p* value < 0.05 was considered statistically significant. Statistical analysis was performed using the program released in 2023 IBM SPSS Statistics for Windows, version 29.0.1.1; IBM Corp, Armonk, NY, USA; and R package (R Core Team, version 4.3.1, 2023).

## 3. Results

Thirty-seven patients were included. Patients’ characteristics are shown in Table 1. According to the LFI, four patients were frail, twenty-eight were prefrail, and five were robust. At baseline, we observed a correlation between LFI and age, muscle mass, and phase angle (Figure 1), but not with the degree of liver insufficiency evaluated by the Child–Pugh and MELD scores. Also at baseline, LFI correlated with LBP, sCD63, mtDNA, and urinary claudin-3 (Figure 2). LFI also showed a statistically significant correlation with uNGAL (r = 0.40, *p* = 0.049) and FGF-21 (r = 0.38, *p* = 0.028). LBP showed a correlation with sCD163 (r = 0.75, *p* < 0.001), sMR (r = 0.70, *p* < 0.001) and mtDNA (r = −0.67, *p* < 0.001).

Regarding metabolomics, we identified 49 metabolites (Figure 3). We observed at baseline a correlation between LFI and mannose, ethanol, and isoleucine (Figure 4). Multivariable linear regression analysis showed that phase angle (coeff. −0.160, 95%CI −0.275; −0.044, *p* = 0.009), sCD163 (coeff. 0.126, 95%CI 0.022; 0.230, *p* = 0.02), FGF-21 (coeff. 0.001, 95%CI 0.000; 0.001, *p* = 0.01), and mannose (coeff. −24.275, 95%CI −43.353; −5.197, *p* = 0.01) were independent factors associated with LFI.

After baseline evaluation, the 32 patients who were frail or prefrail were randomized to either the multifactorial intervention group (n = 17) or the control group (n = 15). Patients’ characteristics of the two groups are shown in Table 1. All patients in both groups reached the 3-month evaluation, 13 and 14 patients reached 6 months, 10 and 14 patients reached 9 months, and 10 and 12 patients reached 12 months, respectively. The remaining patients voluntarily withdrew from the study and no patient died during follow-up.

During follow-up and compared to the control group, patients in the intervention group showed an improvement in LFI, a decrease in CRP, LBP, sCD163, and ccK18, and a trend for sMR to decrease (Figure 5 and Appendix A). No statistically significant changes were observed in muscle mass and phase angle [19], the remaining biomarkers (Appendix A), liver function tests (serum bilirubin, albumin, INR, AST, and ALT), or Child–Pugh and MELD scores [19]. The main changes in metabolomics (Figure 6 and Figure 7 and Appendix A) were a decrease in dimethyl sulfone and creatinine, and an increase in malonate, ornithine, isoleucine and valine in patients in the intervention group in comparison to patients in the control group. We also observed a non-significant trend to an increase in leucine (*p* = 0.06) in the intervention group (Figure 7d).

## 4. Discussion

The main finding of the present study in patients with cirrhosis or ACLD undergoing a multifactorial intervention consisting of home exercise, BCAA, and probiotics was the relationship between frailty and parameters of systemic inflammation, bacterial translocation, and amino acid and short-chain fatty acid metabolism.

Regarding demographic and clinical parameters, we observed that frailty was associated with age and body composition (Figure 1), but not with the degree of liver insufficiency, as previously reported in outpatients with cirrhosis with preserved liver function like patients included in the present study [6]. Interestingly, although weak, we found a correlation between frailty and uNGAL, a relevant predictor of mortality in patients with cirrhosis [23], and FGF-21, a biomarker associated with sarcopenia in these patients [21].

Systemic inflammation plays a key role in pathophysiology and prognosis in cirrhosis, or ACLD [27,28]. Inflammation has also been related to the frailty syndrome [29] in patients with cirrhosis [21] and in other populations [30]. The correlation we observed between LFI and the inflammatory biomarker sCD163 at baseline (Figure 2) and the improvement in CRP and sCD163 and a trend to improve in sMR in parallel with the decrease in frailty during the multifactorial intervention (Figure 5), support the contribution of inflammation to frailty in ACLD. Both sCD163 and sMR are considered biomarkers of macrophage activation and inflammation in liver diseases, and sCD163 is associated with adverse outcomes in patients with cirrhosis [31,32]. sCD163 has also been related to frailty in HIV patients [33]. The mechanisms underlying the relationship between inflammation and frailty may include the aggravation of anorexia, and therefore malnutrition, metabolic disturbances, hyperammonemia, and increased synthesis of myokines [34]. The decrease in inflammatory biomarkers that we observed in patients undergoing the multifactorial intervention could be related to the immunomodulatory effects of exercise [35] and the multistrain probiotic [17].

Pathological bacterial translocation due to gut dysbiosis and alterations in the intestinal barrier is a phenomenon considered to contribute to systemic inflammation in cirrhosis [36]. The correlations that we observed between LBP, an index of bacterial translocation [37], and the parameters of macrophage activation sCD163 and sMR support this relationship. We found a baseline correlation between LFI and LBP (Figure 2) and decreased LBP (Figure 5) after a multifactorial intervention with the potential to modulate gut microbiota and therefore decrease pathological bacterial translocation. Moreover, the correlation between LFI and urinary claudin-3 could indicate a link between frailty and intestinal permeability [17]. These results suggest that bacterial translocation also plays a role in the progression of frailty syndrome in ACLD.

LFI (Figure 2) and LBP also correlated with mtDNA. As frailty has been associated with lower mtDNA levels [38], our results could support the hypothesis of a relationship between frailty, bacterial translocation, and mitochondrial dysfunction; perhaps through systemic inflammation. The decrease observed in ccK18 during the multifactorial intervention (Figure 5) could be due to an improvement in hepatocyte damage and death [39]. However, liver function tests and the degree of liver insufficiency evaluated by the Child–Pugh and MELD scores did not change significantly. It should be noted, nevertheless, that patients presented a relatively well-preserved liver function at baseline and during the study.

Regarding metabolomics, LFI at baseline correlated negatively with mannose and isoleucine and positively with ethanol (Figure 4). Mannose is a carbohydrate whose levels have been related to muscle mass evaluated by dual X-ray absorptiometry in women [40]. Although mannose did not correlate with muscle mass in our study, the correlation between this carbohydrate and LFI, a composite index in which muscular function—evaluated by handgrip strength and chair stands—plays a key role, suggests a relationship between mannose and muscular health. Isoleucine is a BCAA whose low levels have been associated with frailty in elderly patients [41]. Ethanol has been associated with frailty in the general population [42] and sarcopenia in cirrhosis [21]. These observations suggest that frailty in patients with ACLD is related to carbohydrate and amino acid metabolism and a possible surreptitious intake of alcohol or alcohol overproduction by dysbiotic microbiota [43].

During the multifactorial intervention, metabolomics showed a decrease in dimethyl sulfone and creatinine and an increase in malonate, ornithine, isoleucine, and valine, and a non-significant trend for leucine to increase (Figure 6 and Figure 7). Dimethyl sulfone, a sulfur-containing molecule with antioxidant properties [44,45], has been shown to be increased in patients with a high risk of diabetic retinopathy [46] and decreased in those persons following a healthy diet [47]. The meaning of the decrease in dimethyl sulfone observed in the present study is not clear. It could be related to a lower need for a scavenging effect on reacting oxygen species as a consequence of the decrease in bacterial translocation and inflammation [48], or to the exhaustion of this biomarker as a result of an increase in oxidative stress. This second possibility seems unlikely considering the positive effects we observed in clinical parameters [19] and other biomarkers. Whatever the case, the lack of significant changes in serum MDA results (Appendix A) does not support the hypothesis of a significant change in oxidative damage. Another potential explanation for the decrease in dimethyl sulfone is a change in the microbiota because this metabolite is produced by gut bacteria [49], and the multifactorial intervention used here indeed has the potential for microbiota modulation [50,51]. Unfortunately, we did not analyze the changes in gut microbiota composition in the present study.

The decrease in dimethyl sulfone and creatinine and the increase in valine observed in the present study could reflect an improvement in renal function [52] in the context of a decrease in inflammation. In effect, as reported in our previous article [19], patients in the intervention group presented a non-significant decrease in serum creatinine measured by a routine automated Jaffé method. In the present study, using ^1^H-NMR with correction for water content the improvement in creatinine reached statistical significance. This difference is most likely due to the different methods used. The decrease in creatinine could also be related to a decrease in muscle mass, but this seems unlikely because we did not observe significant changes in muscle mass evaluated by BIA. In spite of these findings, we observed an improvement in muscle function after the multifactorial intervention [19].

Regarding the increase in BCAA—isoleucine and valine, and a non-significant trend for leucine—this could have been expected because patients in the intervention group received these amino acids. A decrease in BCAA is a well-recognized feature in patients with ACLD, related to hepatic encephalopathy and worse prognosis [13,53]. Treatment with BCAA has been recommended in these patients to improve sarcopenia and cognitive function and to prevent hepatic encephalopathy [13]. The increase in serum BCAA observed in our study could explain, at least in part, the positive effects of the multifactorial intervention on frailty and, as previously reported, on clinical events [19]. Ornithine is another amino acid involved in the metabolism of ammonia. It has been combined with aspartate to form L-ornithine-L-aspartate (LOLA), a molecule widely used to treat hyperammonemia and hepatic encephalopathy in cirrhosis [54]. Its increase might suggest a higher capacity of ammonia detoxification through the urea cycle.

Malonate is a short-chain fatty acid generated by gut microbiota from dietary fiber. It could therefore change after the intervention that targeted gut microbiota in our study. On the other hand, it has been suggested that malonate can induce beneficial metabolic effects, such as modulation of mitochondrial function, increased oxygen consumption, and browning of white adipocytes [55].

Our study has several limitations, the first of these being the relatively small sample size. However, the samples come from patients included in a randomized controlled trial and closely followed every 3 months for a relatively long period of time. Second, the patients were in a stable condition and presented a relatively well-preserved liver function. Perhaps the results would differ in patients with more advanced liver disease. Finally, due to the multifactorial nature of the intervention, which included home exercise, BCAA supplementation, and a multistrain probiotic, we cannot appraise the specific contribution of each of these treatments to the effects reported herein. Nonetheless, we believe our results provide valuable information regarding the possible mechanisms underlying frailty and its improvement after a non-pharmacological multifactorial intervention in patients with ACLD and preserved liver function before the disease progresses further.

## 5. Conclusions

We conclude that frailty in patients with ACLD is associated with parameters of systemic inflammation, bacterial translocation and liver damage, and alterations in the metabolism of amino acids and short-chain fatty acids. These results may help identify new therapeutic targets aiming to improve frailty and its consequences in patients with ACLD.

## Figures and Tables

**Figure 1 biomolecules-14-01410-f001:**
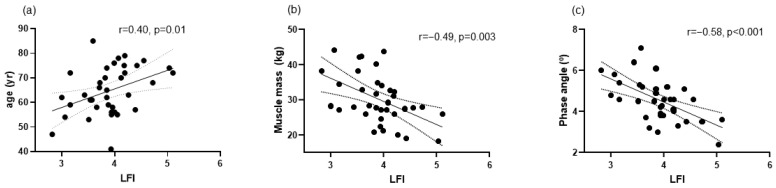
Correlations at baseline between liver frailty index (LFI) and: (**a**) age, (**b**) muscle mass, and (**c**) phase angle assessed by bioelectrical impedance analysis (BIA).

**Figure 2 biomolecules-14-01410-f002:**
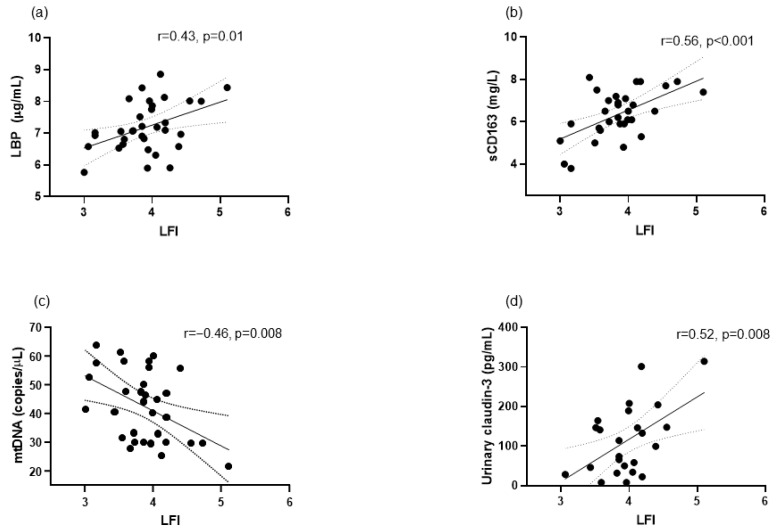
Correlations at baseline between liver frailty index (LFI) and: (**a**) lipopolysaccharide binding protein (LBP), (**b**) soluble CD163 (sCD163), (**c**) mitochondrial DNA (mtDNA), and (**d**) urinary claudin-3. LBP was available in 33 patients, sCD163 in 31 patients, mtDNA in 32 patients, and urinary claudin-3 in 25 patients.

**Figure 3 biomolecules-14-01410-f003:**
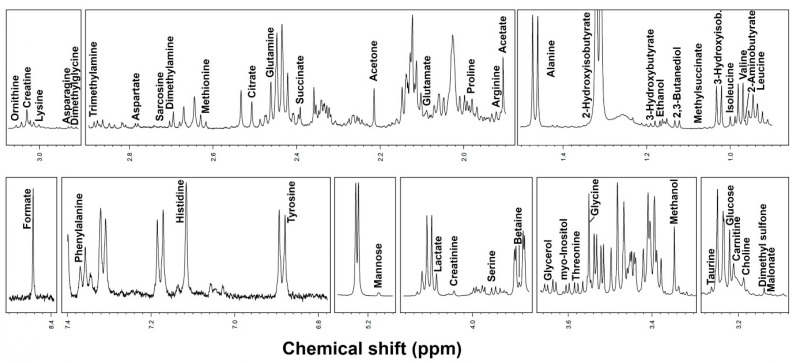
Parts of spectra obtained by ^1^H-NMR, representative of all the spectra obtained in the present study. The name of each molecule appears over the NMR signal used for its quantification. To ease the visual of each portion, a different vertical magnification has been selected.

**Figure 4 biomolecules-14-01410-f004:**
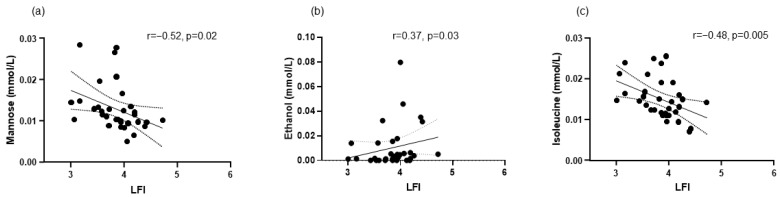
Correlations at baseline between liver frailty index (LFI) and the metabolites: (**a**) mannose, (**b**) ethanol, and (**c**) isoleucine. Mannose, ethanol, and isoleucine were available in 33 patients.

**Figure 5 biomolecules-14-01410-f005:**
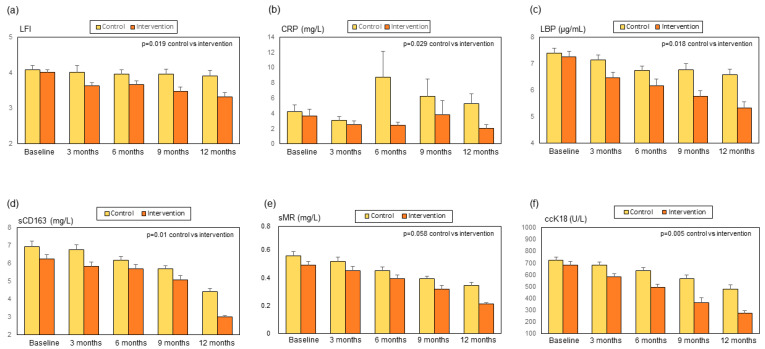
Changes during the study in the control group and the intervention group: (**a**) liver frailty index (LFI), (**b**) c-reactive protein (CRP), (**c**) lipopolysaccharide binding protein (LBP), (**d**) soluble CD163 (sCD163), (**e**) soluble mannose receptor (sMR), and (**f**) caspase-cleaved keratin 18 (ccK18).

**Figure 6 biomolecules-14-01410-f006:**
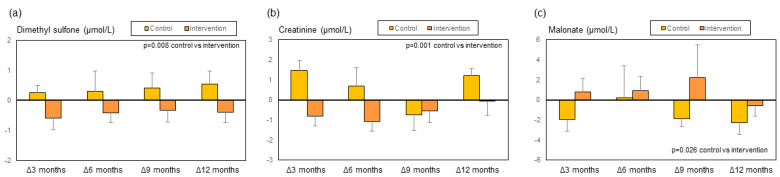
Changes during the study in the metabolites in the control group and the intervention group expressed as delta change with respect to baseline values: (**a**) dimethyl sulfone, (**b**) creatinine, and (**c**) malonate.

**Figure 7 biomolecules-14-01410-f007:**
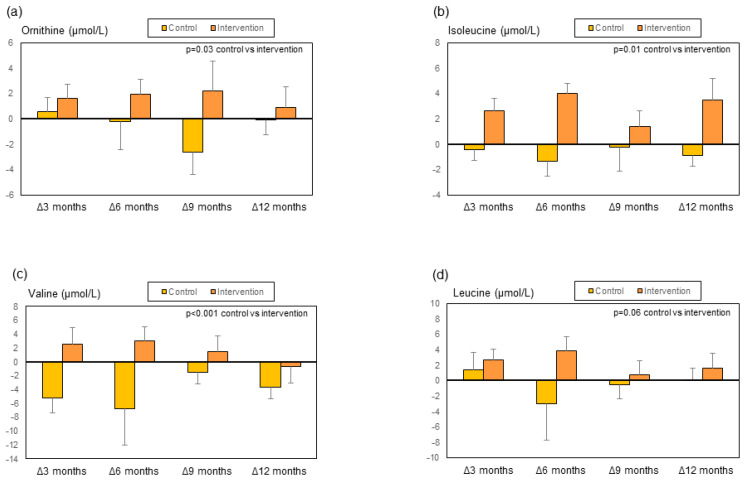
Changes during the study in the metabolites in the control group and the intervention group expressed as delta change with respect to baseline values: (**a**) ornithine, (**b**) isoleucine, (**c**) valine, and (**d**) leucine.

**Table 1 biomolecules-14-01410-t001:** Baseline characteristics of all patients, patients who are robust, and patients who are prefrail or frail in the control group and the intervention group. Numbers in bold indicate statistically significant differences.

	All Patients (n = 37)	Robust Patients (n = 5)	Prefrail and Frail Patients (n = 32)	
			Control Group (n = 15)	Intervention Group (n = 17)
Age (yr)	64.7 ± 1.6	58.8 ± 4.1	68.1 ± 2.4	63.5 ± 1.7
Male/female (%)	23 (62)/14 (38)	4 (80)/1 (20)	8 (53.3)/7 (46.7)	11 (64.7)/6 (35.3)
Etiology: alcohol/virus/MASLD/other (%)	26 (70)/3 (8)/ 4 (11)/4 (11)	4 (80)/0/1 (20)/0	9 (60)/1 (6.6)/ 2 (13.3)/3 (20)	13 (76.5)/2 (11.7)/ 1 (5.8)/1 (5.8)
Previous decompensations (%)	30 (81)	4 (80)	12 (80)	14 (82.4)
Child–Pugh score	5.5 ± 0.2	5.8 ± 0.8	5.6 ± 0.3	5.3 ± 0.2
MELD score	8.7 ± 0.5	10.6 ± 2.6	8.5 ± 0.4	7.7 ± 0.4
Comorbidity index (Charlson)	5.8 ± 0.3	4.6 ± 0.9	6.3 ± 0.4	5.7 ± 0.3
BMI (kg/m^2^)	28.0 ± 0.6	27.8 ± 1.8	28.9 ± 0.8	27.5 ± 1.1
LFI	3.90 ± 0.08	**3.04 ± 0.06 ^1^**	4.07 ± 0.12	4.00 ± 0.08
Phase angle (°)	4.5 ± 0.2	5.3 ± 0.3	4.5 ± 0.3	4.3 ± 0.2
Creatinine (µmol/L)	72.5 ± 2.1	77.8 ± 5.9	71.2 ± 3.3	72.1 ± 3.2
Bilirubin (µmol/l)	21.5 ± 2.9	22.6 ± 3.2	22.7 ± 4.3	17.4 ± 1.9
Albumin (g/L)	37.9 ± 0.9	39.8 ± 2.9	37.4 ± 1.2	37.5 ± 1.5
INR	1.13 ± 0.04	1.19 ± 0.12	1.14 ± 0.04	**1.05 ± 0.02 ^2^**

MASLD: metabolic dysfunction-associated steatotic liver disease; MELD: model for end-stage liver disease; BMI: body mass index; LFI: liver frailty index; INR: international normalized ratio. ^1^
*p* < 0.001 with respect to control group and intervention group; and ^2^ *p* = 0.04 with respect to control group. No statistically significant differences were observed between the three groups in the remaining parameters.

## Data Availability

The raw data supporting the conclusions of this article will be made available by the authors on request.

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
