# Peer review of "Biomarkers of Frailty in Patients with Advanced Chronic Liver Disease Undergoing a Multifactorial Intervention Consisting of Home Exercise, Branched-Chain Amino Acids, and Probiotics"

_biomolecules, 2024, doi:10.3390/biom14111410_

Round 1
Reviewer 1 Report
Comments and Suggestions for Authors
INTRODUCTION:
The introduction is too brief and should be expanded. I suggest the following points:
- You could introduce the concept that, in liver disease patients, there is an emerging trend of using nutritional supplementation with bicarbonate-magnesium-rich water, low in sodium. I would cite studies on Fonte Essenziale water, which has also shown improvements in gut flora and IBS-like symptoms.
- I would also mention that patients with gastrointestinal tract diseases tend to have a reduced propensity for physical activity (for example, you could reference the two BE-FIT-IBD studies).
OTHER:
- I would avoid using the term cirrhosis, and instead prefer cACLD or dACLD where applicable.
- Improve the resolution of Figures 5, 6, and 7.
Author Response
Thank you for your interesting comments. We have done our best to answer your queries.
The introduction is too brief and should be expanded. I suggest the following points:
- You could introduce the concept that, in liver disease patients, there is an emerging trend of using nutritional supplementation with bicarbonate-magnesium-rich water, low in sodium. I would cite studies on Fonte Essenziale water, which has also shown improvements in gut flora and IBS-like symptoms.
- I would also mention that patients with gastrointestinal tract diseases tend to have a reduced propensity for physical activity (for example, you could reference the two BE-FIT-IBD studies).
Response. Thank you for these interesting comments. We agree that the Introduction was too brief and so we have now included additional information regarding the frailty syndrome in general and in association with chronic diseases (pages 1 and 2, lines 40-48). We have also added a comment and a reference about Fonte Essenziale as a non-pharmacological potential treatment to improve frailty in liver diseases, and a sentence regarding the low physical activity in patients with chronic diseases (we cite the first BE-FIT-IBD study), including cirrhosis or advanced chronic liver disease (ACLD) (page 2, lines 56-59). As a consequence, we have added 5 new references.
OTHER:
- I would avoid using the term cirrhosis, and instead prefer cACLD or dACLD where applicable.
Response. We agree that in general the term ACLD is preferable to cirrhosis. Most of our patients (81%) had previous decompensations (dACLD) (and probably cirrhosis) and only 19% did not (cACLD). In any case, we have now changed the word “cirrhosis” for “ACLD” in the title and in most places in the manuscript.
- Improve the resolution of Figures 5, 6, and 7.
Response. We have now improved the resolution of Figures 5, 6, and 7.
Reviewer 2 Report
Comments and Suggestions for Authors
Ladhi L et al's manuscript shows biomarkers of frailty in patients with cirrhosis undergoing exercise, BCAA and probiotics. The authors analyzed comprehensive metabolomics of blood samples from outpatients with high LFI values using 1H-NMR, and extracted biomarkers related to frailty control in patients with liver cirrhosis.
Major comments
Can the authors put more one table explaining the clinical mean of each biomarker?
In Discussion section, show the Figure No. associated with contents.
In matobolomics, the multifactorial intervention decreased dimethyl sulfone and creatinine. Page 9 has the discussion of this result. What is the source of dimethyl sulfone, and how is dimethyl sulfone created? Is dimethyl sulfone good for health or not? In Page 9, the authors mentioned the modulation of gut microbiota, but the multifactorial intervention included probiotics here. Please discuss the decrease in dimethyl sulfone in detail. And why did creatinine decrease, even though the outpatients exercise?
Next paragraph, the authors discussed the hepatic encephalopathy concerned with increase in BCAA. Among amino acids other than BCAA, Pro, Lys and Asn also displayed an increasing tendency. The cirrhosis outpatients with the multifactorial intervention are thought to be at high risk for hepatic encephalopathy. What do the authors think?
Minor comments
Page 6, Line6 "Supplementary TableS1", please add "S".
Line15 Add "(Figure 7D)" in the last of sentence.
Author Response
Thank you for your thoughtful comments. We have done our best to try to answer your questions.
Laghi L et al's manuscript shows biomarkers of frailty in patients with cirrhosis undergoing exercise, BCAA and probiotics. The authors analyzed comprehensive metabolomics of blood samples from outpatients with high LFI values using 1H-NMR, and extracted biomarkers related to frailty control in patients with liver cirrhosis.
Major comments
Can the authors put more one table explaining the clinical mean of each biomarker?
Response. Many thanks for your comments. We have now added a new table (Supplementary Table S1) with the clinical meaning of the battery of biomarkers. We agree this can help to understand the clinical implications of the results.
In Discussion section, show the Figure No. associated with contents.
Response. We have added the figure number in parentheses alongside the comments related to each figure.
In metobolomics, the multifactorial intervention decreased dimethyl sulfone and creatinine. Page 9 has the discussion of this result. What is the source of dimethyl sulfone, and how is dimethyl sulfone created? Is dimethyl sulfone good for health or not? In Page 9, the authors mentioned the modulation of gut microbiota, but the multifactorial intervention included probiotics here. Please discuss the decrease in dimethyl sulfone in detail. And why did creatinine decrease, even though the outpatients exercise?
Response. We do not know the exact meaning of the decrease in dimethyl sulfone and whether or not it is good for health. However, we suppose that in the context of our study it is good for health. We have found in the literature that dimethyl sulfone is a sulfur containing molecule with antioxidant properties and that it is produced by gut bacteria. It has been used as an antioxidant, and it has shown to be increased in patients at high risk of diabetic retinopathy and decreased in those persons following a healthy diet. We do not know if the decrease in dimethyl sulfone in our patients means an improvement in oxidative stress, and thus a lower need for a scavenging effect on reacting oxygen species, or whether it means exhaustion because of an increase in oxidative stress. The possibility of exhaustion of this biomarker as the result of an increase in oxidative stress seems unlikely considering the positive effects we observed in clinical parameters and other biomarkers. We have now added the results of blood malondialdehyde (MDA), a biomarker of oxidative stress (Supplementary Table S2), and MDA did not change during the study. This lack of significant changes in MDA does not support the hypothesis of a change in oxidative stress. Another potential explanation for the decrease in dimethyl sulfone is a change in the microbiota because dimethyl sulfone is produced by gut bacteria. In effect, the multifactorial intervention used here included probiotics, and therefore has the potential to modulate gut microbiota. Unfortunately, in this study we did not analyse the changes in gut microbiota composition after the multifactorial intervention.
We have tried to explain better these aspects in the Discussion (page 9, lines 308-319)).
Regarding the decrease in creatinine, we consider this might be related to an improvement in renal function or to a decrease in muscle mass. Although patients in the intervention group showed an improvement in muscle function, we did not observe significant changes in muscle mass evaluated by bioelectrical impedance analysis (BIA). Therefore, the multifactorial intervention improved muscle function but did not produce significant changes in muscle mass. Hence, we consider the most probable hypothesis to explain the decrease in creatinine is an improvement in renal function in the context of a decrease in inflammation.
We have tried to improve the explanation of the change in creatinine in the Discussion (pages 9 and 10, lines 320-330).
Next paragraph, the authors discussed the hepatic encephalopathy concerned with increase in BCAA. Among amino acids other than BCAA, Pro, Lys and Asn also displayed an increasing tendency. The cirrhosis outpatients with the multifactorial intervention are thought to be at high risk for hepatic encephalopathy. What do the authors think?
Response. Thank you for this comment. Patients with cirrhosis usually present a decrease in BCAA that has been related to the development of hepatic encephalopathy and a worse prognosis. Thus, treatment with BCAA has been recommended in patients with cirrhosis or advanced chronic liver disease (ACLD) to improve sarcopenia and cognitive function, and to prevent hepatic encephalopathy. Therefore, the increase in BCAA observed in the patients treated with the multifactorial intervention should be considered beneficial. We have tried to clarify this in the Discussion (page 10, lines 334-337).
Minor comments
Page 6, Line6 "Supplementary TableS1", please add "S".
Response. Done.
Line15 Add "(Figure 7D)" in the last of sentence.
Response. Done.